# Identification of the Antigens Recognised by Colorectal Cancer Patients Using Sera from Patients Who Exhibit a Crohn’s-like Lymphoid Reaction

**DOI:** 10.3390/biom12081058

**Published:** 2022-07-29

**Authors:** Viktoriya B. Boncheva, Michael Linnebacher, Said Kdimati, Hannah Draper, Laurence Orchard, Ken I. Mills, Gerald O’Sullivan, Mark Tangney, Barbara-ann Guinn

**Affiliations:** 1Department of Life Sciences, University of Bedfordshire, Park Square, Luton LU1 3JU, UK; viktoriya.boncheva@outlook.com; 2Department of General Surgery, Molecular Oncology and Immunotherapy, University Medical Center Rostock, Schillingallee 69, 18057 Rostock, Germany; michael.linnebacher@med.uni-rostock.de (M.L.); said.kdimati@med.uni-rostock.de (S.K.); 3Hull York Medical School, University of Hull, Cottingham Road, Kingston-upon-Hull HU6 7RX, UK; hannah.draper-2021@hull.ac.uk; 4Cancer Sciences, Southampton University Hospitals Trust, University of Southampton, Southampton SO16 6YD, UK; l.orchard88@gmail.com; 5Patrick G. Johnson Centre for Cancer Research, Queen’s University Belfast, Lisburn Road, Belfast BT9 7AE, UK; k.mills@qub.ac.uk; 6Cancer Research@UCC, Cork, Ireland; m.tangney@ucc.ie; 7SynBioCentre, University College Cork, Cork, Ireland; 8APC Microbiome Ireland, University College Cork, Cork, Ireland; 9Department of Biomedical Sciences, University of Hull, Cottingham Road, Kingston-upon-Hull HU6 7RX, UK

**Keywords:** colon cancer, SEREX, crohn’s-like lymphoid reaction (CLR), immunotherapy, immunoglobulin heavy chain

## Abstract

A Crohn’s-like lymphoid reaction (CLR) is observed in about 15% of colorectal cancer (CRC) patients and is associated with favourable outcomes. To identify the immune targets recognised by CRC CLR patient sera, we immunoscreened a testes cDNA library with sera from three patients. Immunoscreening of the 18 antigens identified by SEREX with sera from normal donors showed that only the heavy chain of IgG3 (IGHG3) and a novel antigen we named UOB-COL-7, were solely recognised by sera from CRC CLR patients. ELISA showed an elevation in IgG3 levels in patients with CRC (*p* = 0.01). To extend our studies we analysed the expression of our SEREX-identified antigens using the RNA-sequencing dataset (GSE5206). We found that the transcript levels of multiple IGHG probesets were highly significant (*p* < 0.001) in their association with clinical features of CRC while above median levels of DAPK1 (*p* = 0.005) and below median levels of GTF2H5 (*p* = 0.004) and SH3RF2 (*p* = 0.02) were associated with improved overall survival. Our findings demonstrate the potential of SEREX-identified CRC CLR antigens to act as biomarkers for CRC and provide a rationale for their further characterization and validation.

## 1. Introduction

Colorectal cancer (CRC) is one of the most commonly diagnosed cancers worldwide. It affects the bowel and rectum, is rare in people under 40, with almost 85 per cent of cases being diagnosed in persons over 65 years of age. One in every twenty people in the UK develops CRC, with only half of them surviving beyond five years, mainly because CRC is often detected once well-established and after it has spread beyond the bowel. The disease stage at the time of diagnosis governs both the choice of treatment and the prognosis. The most commonly used staging system for CRC is the American Joint Committee on Cancer (AJCC), also known as the tumour, node, metastases (TNM) system [1], which reflects how far the cancer has spread, and whether it has reached nearby structures such as lymph nodes and/or distant organs.

Stage I CRC is an asymptomatic malignancy, developing slowly through the progressive accumulation of genetic mutations within precancerous bowel lesions and polyps. Diagnosis at this stage reduces the risk of death from CRC, giving a 90% chance of survival beyond five years [2] and a significantly decreased frequency of disease recurrence. However, most cases of CRC are detected once the cancerous cells have moved beyond the middle layers of the colon (Stage II). Treatment generally involves surgery, chemotherapy and/or radiotherapy, with little in the way of immunotherapy except for the approximately 15% of CRCs that show mismatch repair deficiency or microsatellite instability (MSI). In these patients, there have been promising results with immune checkpoint inhibitor (ICI) treatment, utilising pembrolizumab and nivolumab to block programmed cell death 1, resulting in improved survival, even in metastatic CRC (reviewed in [3]). However, the vast majority of CRC patients have been shown to be largely unresponsive to ICI treatment. Current research focusses on how to improve the immune response to ICIs in patients that are microsatellite stable/DNA mismatch repair proficient (reviewed in [4]). Currently, groups are examining ways to make microsatellite stable CRC an “immune hot” tumour through the examination of components of the immune system as well as studies of the potential of bispecific antibodies, cellular therapies, vaccines and cytokines to stimulate an anti-tumour immune response.

The Crohn’s-like reaction (CLR) is a CRC specific ectopic lymphoid reaction characterised by peritumoural aggregates at the advancing edge of the tumour (reviewed in [5]). Early in CLR development CD4+ T cells cluster mostly with mature dendritic cells but as the CLR matures, B cell numbers increase and follicular dendritic cells are recruited and create lymphoid follicles. CLR CRC was first described in 1990 [6] and is associated with improved survival among CRC patients. It should be noted that CLR has no biological relationship to Crohn’s disease and is in fact a CRC-specific ectopic or tertiary lymphoid reaction, with similar structures found in other non-CRC tumours. CRC tumours deficient in the DNA mismatch repair system have MSI and a high mutational burden associated with the presence and density of CLR but it is solely the subgroup of CRC known to have CLRs that we have focused on in this study.

With the aim of trying to find new immune targets for CRC therapy, we wanted to determine which antigens were being recognised by sera from patients with CLR-CRC, who showed an improved overall survival and were able to clear post-operative micrometastasis [7]. We focused on this group in the hope of identifying antigens that could be used as targets for CRC immunotherapy in the future.

## 2. Materials and Methods

### 2.1. Patient and Healthy Volunteer (HV) Serum Samples

Diagnostic and clinical investigations for staging all CRC patients who attended the Mercy Hospital Cork were performed in accordance with the guidelines of the human ethics committee for clinical research, National University of Ireland, Cork. Histological examination showed tumour extension beyond the muscularis propria and absence of nodal metastases. Three Dukes’ B non-synchronous CRC patients who had no concurrent inflammatory bowel disease or familial polyposis coli had been treated surgically with radical resection of the primary tumour including mesentary of the bowel in a 3-month period during 2007 and were followed up for micrometastases in a dedicated clinic for a mean period of 3 years post-surgery. CC005 (male, aged 67), CC010 (male, aged 77) and CC014 (female, aged 88) each met the criteria described by Murphy et al. 2000 [7] as having Jass and “Crohn’- like” lymphoid reactions during follow-up appointments. The other 12 Dukes’ B/stage II CRC samples collected during this same period in 2007 did not meet the CLR criteria and, thus, were not used for immunoscreening in our study but were used in ELISAs.

HV blood samples were obtained from The National Blood Service, North London Blood Transfusion Centre and used for immunoscreening. Serum was collected from clotted peripheral blood samples following centrifugation at 1200× *g* rpm for 10 min, aliquoted and frozen at −80 °C until required. The testes library was prepared from cDNAs from five HVs aged between 19 and 65 (Stratagene Europe, Amsterdam, The Netherlands) [8].

The clinical and pathological characteristics of the CRC samples used to examine IgG3 levels by immunohistochemistry (IHC) are summarised in Appendix A. The process of establishing patient-derived cell lines has been reported previously [9], and primary cell lines, directly established from human fresh tumour tissues, are indicated with the prefix HROC and the ID number of the patient, e.g., HROC60. All patients signed the written informed consent and the procedures were approved by the Ethics Committee of the University Hospital of Rostock (reference number II HV 43/2004 and A 45/2007) in accordance with the declaration of Helsinki.

### 2.2. Immunoscreening of the Testes cDNA Library and Identification of Positive Clones

Serological analysis of recombinant cDNA expression libraries (SEREX) was performed as described previously [10] with minor modifications [11]. Briefly, the testes cDNA library was prepared following the manufacturer’s instructions [8] from five normal donors aged between 19 and 65 (Stratagene Europe, Amsterdam, The Netherlands). Phage containing cDNA inserts were plated at 10^4^ pfu per 132 mm Petri dish onto a lawn of XL1 Blue MRF’ *E. coli.* Sera was cleared using *E. coli* XL1 Blue MRF’ lysates, with and without phage lacking a cDNA insert, either bound to CNBr activated sepharose 4B beads (Amersham Pharmacia Biotech, Buckinghamshire, UK) or 132 mm 0.2 µm circular nitrocellulose membranes (PALL Corporation, New York, NY, USA). The reactivity of serum was confirmed through agglutination tests [12] using 30 µL sheep and/or pigs blood and 15 µL CRC sera diluted 1:10 in phosphate-buffered saline. Pre-cleared CRC sera were diluted in TBS/0.05% sodium azide and used for primary and secondary immunoscreening (the latter to confirm sero-positivity). All positive plaques were isolated to monoclonality, eluted, in vivo excised (following Stratagenes ‘ZAP Express cDNA Synthesis kit’ manufacturers’ guidelines, Cat #200451) and plated as phagemids in *E. coli* on selective media plates. Single colonies were expanded by overnight culture and plasmid DNA was isolated using a Qiagen plasmid mini kit (Qiagen Ltd., Sussex, UK, Cat #12123).

### 2.3. Sequence Analysis of cDNA Inserts

T3 and T7 primers (5′-GCAATTAACCCTCACTAAAGG-3′ and 5′-TAATACGACTCACTATAGGG-3′, respectively) bind regions that flank the multiple cloning site. They were used to amplify the cDNA inserts in the pBK-CMV phagemid of immunoresponsive clones with ReadyMix Taq PCR Reaction Mix with MgCl_2_ (Sigma Aldrich, UK; Cat #P4600). PCR products were gel purified using a QIAQuick gel extraction kit (Qiagen Ltd., Manchester, YORK, UK; Cat #28704) and sent to the Department of Biochemistry at the University of Cambridge [13] for Sanger sequencing. Nucleotide sequences generated by Applied Biosystems’ Sequencing Analysis Software and predicted amino acid sequences were compared with known sequences in the gene, expressed sequence tag (EST) and protein databases, including the National Centre for Biotechnology Information (NCBI) BLAST, Baylor College of Medicine (BCM) Search launcher, the U.K. human genome mapping project (hgmp) resource centre and The Institute for Genomic Research (TIGR) web-based facilities.

### 2.4. Immunoscreening of Testes Responsive Clones with Sera from Immune-Responsive CRC Patients and HVs

Every unique antigen (named UOB-COL-1-18) was immunoscreened with sera from each CRC patient with CLR (CC005, CC010 and CC014), while UOB-COL-1 to -15 were also immunoscreened with eight age- and sex-matched HV sera.

### 2.5. Microarray Analysis

Utilising existing microarray data, chosen by virtue of their sample size, availability of clinical details and suitability of the controls, we compared gene transcript levels to clinical features, as well as above and below median expression and its relationship to overall survival. The GSE5206 dataset [14] was generated from the analyses of 100 human CRC samples encompassing all clinical stages [15] and five normal colon tissues. We examined the levels of expression of each antigen in CRC patients versus normal donors and any correlation between levels of expression and clinical features. We additionally analysed IGHG2 and IGHG3 gene expression using The Cancer Genome Atlas (TCGA) Colon Adenomarcinoma (COAD) dataset [16] which assesses 571 CRC patients by RNA-sequencing. This dataset is publicly available through xenabrowser.net and the dataset was chosen because it is one of the largest and most annotated datasets available.

### 2.6. IHC

Immunolabelling of IgG3 was performed using 3.5 μg/mL monoclonal rabbit anti-human IgG3 (clone RM119, Sigma-Aldrich, Cat# SAB5600212, St. Louis, MO, USA) with a human tonsil as a positive control. Clone RM119 reacts to the heavy chain of IgG3 and not to any other IgG subclasses (IgG1, IgG2 or IgG4), nor does it show any cross reactivity with IgM, IgA, IgD or IgE. IgG antibody was also used as a control for the presence of other IgG subclasses. Immunolabelling was detected using the EnVision^®^+ Dual link system (DAB+), HRP-labelled anti-rabbit polymere (Agilent Technologies Deutschland GmbH, Waldbronn, Germany; Cat#K4063), following the manufacturer’s instructions. Blocking and dilution buffers contained 2% bovine serum albumin (Sigma-Aldrich, Cat#A9418).

### 2.7. ELISA

We detected IgG3 levels in sera from 12 CRC patients (CC001-CC012) and 12 normal donors using a pre-coated solid-phase sandwich ELISA kit IgG3 (Invitrogen Cat#BMS2094, Waltham, MA, USA) and following the manufacturer’s instructions.

## 3. Results

### 3.1. Eighteen Antigens Were Recognised by CLR-CRC Patient Sera

Testes have many features similar to cancer, including rapid proliferation and global promoter hypomethylations, offering an opportunity to identify cancer-testis antigens (recently reviewed in [17]) and immunoscreen a broad range of proteins [18]. Primary immunoscreening was performed following the SEREX technique on 821,644 plaque forming units (pfu) using either CC005 (1:2000 and 1:200 dilutions), CC010 (1:200 dilution) or CC014 (1:200 dilution) serum. Each of these patients had been identified as meeting the criteria for CLR-CRC as described by Murphy et al. 2000 [7]. A total of154 phages were identified as potentially sero-positive and secondary immunoscreening confirmed 32 phages harboured immune reactive cDNA inserts using the same sera that they were identified with.

Following amplification of cDNA inserts in the pBK-CMV vector using T3-T7 primers, products were isolated from 1% agarose gel using QIAquick gel extraction and Nanodrop analysis performed to quantify the cDNA. Agarose gel electrophoresis demonstrated that a range of cDNA insert sizes had been sero-recognised by CRC CLR sera in the testes cDNA library (data not shown), indicating the cDNA library were representative of a breadth of cDNA transcripts (as shown in [8]).

In total, 32 sero-responsive clones that encoded 18 independent antigens (Table 1) were named UOB-COL-1 to UOB-COL-18 following SEREX convention [19]. The nine individual cDNA fragments that were mapped to corresponding genes did so with maximum identity values ranging between 90% and 99%. The differences between the query sequence (UOB-COL cDNA insert) and the subject were single nucleotides and most likely due to errors in the sequencing process. FASTA sequences are already available for these sequences [8].

### 3.2. Two Proteins Were Recognised by CLR-CRC but Not HV Patient Sera

To determine which antigens were solely recognised by CLR-CRC sera, we immunoscreened each unique cDNA with sera from each CLR-CRC patient and eight HVs. Fifteen of the eighteen immunoresponsive sequences were identified and confirmed with serum CC014, nine with CC010 and ten with serum CC005 (Table 2). None of the HV sera reacted against UOB-COL-1, an antigen found to be strongly responsive with all three of the sera from patients with CLR-CRC, or UOB-COL-7. UOB-COL-1 encoded the immunoglobulin heavy constant gamma 3 (G3m Marker) (IGHG3), while the sequence of UOB-COL-7 did not map onto a known gene. Each HV sera reacted with at least six of the antigens and in many cases reacted with up to 12 of the UOB-COL antigens.

### 3.3. Confirmation of Ig Heavy Chain Recognition by Patient Sera

To check the validity of the identification of the IGHG3 and IGHG2 proteins (identified as UOB-COL-1 and 2, respectively, in our study), we immunoscreened part of the testes cDNA library with anti-IgG (secondary) antibody only. No positives were found following the immunoscreening of more than 300 pfu and the same antigens had not been identified during previous immunoscreening studies by our group [20]. This suggests that the heavy chain fragments (IGHG2 and IGHG3) were not identified as a product of contamination but due to recognition of IgG heavy chains by the responsive CRC patient sera. In addition, UOB-COL-1 was not sero-recognised by eight HV sera, suggesting the result was not an artefact of the sero-screening method.

### 3.4. Only IGHG2 and IGHG3 Had Elevated Transcription in CRC Patient Tissue

We examined the expression of the nine known antigens identified by SEREX in this study ((IGHG2, IGHG3, ZNF465, CYB5R3, SLC34A2, RPL37A, SH3RF2, DAPK1 and GTF2H5) using the publicly available RNA-sequencing dataset (GSE5206 [14]). The only probesets with elevated levels (>2 fold or *p* < 0.05) in patients when compared with healthy donor tissues were those that included IGHG2 or IGHG3 (Table 3).

Using an independent dataset (TCGA-COAD) The Cancer Genome Atlas (TCGA) Colon Adenomarcinoma (COAD) dataset [16] which annotates 571 CRC patient samples by RNA-sequencing, we did not observe a unique sub-population of CRC patients with elevated IGHG3 expression in their tumours (Figure 1). However, we did find a significant association between the expression of IGHG2 and IGHG3, suggesting that the transcription of these constant regions have similar expression profiles in CRC patients (*p* = 8.25 × 10^−145^).
biomolecules-12-01058-t001_Table 1Table 1BLAST results for the nine previously identified antigens identified by CRC CLR patients.NameGene SymbolChromosome LocalisationGeneral Function of the Encoded Protein ^#^SEREXIdentified AntigenSignificance in Different Types of CancerUOB-COL-1**IGHG3**14q32.33Involvement in a number of molecular, biological and cell-signalling pathways**No**Potential: Diagnostic marker in malignant mesothelioma [21]; Diagnostic and prognostic marker in prostate cancer [22]; Prognostic marker in breast cancer [23]; overexpressed in non-small cell lung carcinoma [24].UOB-COL-2**IGHG2**14q32.33Involvement in a number of molecular, biological and cell-signalling pathways**No**No known tumour-associated properties.UOB-COL-3**AZFP**2q21.2Novel family of genes with unidentified general function**Yes**Encodes a novel transmembrane zinc finger protein with a KRAB box domain. Found to be overexpressed in a patient with acute myeloid leukaemia and detected with autologous serum, SEREX id: GKT-AML8 [25]. Also found to be overexpressed in CML patients and several cancer cell lines but not in normal donor blood cells.UOB-COL-4**CYB5R3**22q13.2Desaturation and elongation of fatty acids, cholesterol biosynthesis, drug metabolism, and, in erythrocyte, methemoglobin reduction**No**Polymorphism associated with increased breast cancer risk [26]. UOB-COL-5**SLC34A2**4p15.2pH-sensitive sodium-dependent phosphate transporter**No**Downregulation in A549 (lung adenocarcinoma cells) promotes tumour development [27]; Overexpressed in breast cancer and could act as potential therapeutic target [28]; Its encoded protein (NaPi2b) is targeted by Rebmab200 (humanised monoclonal antibody) in cancer [29]; Potential diagnostic marker in ovarian cancer [30].UOB-COL-11**RPL37A**2q35Structural constituent of ribosome**Yes ***Showed to predict response to neoadjuvant doxorubicin and cyclophosphamide in breast cancer patients (as part of a panel of genes) [31].UOB-COL-16**SH3RF2**5q32Promotes cell survival and apoptosis. Inhibits PPP1CA phosphatase activity.**No**Overexpressed in human cancers and regulates PAK4 in colon cancer. Acts as an oncogene and may represent an effective therapeutic target for cancer treatment [32]. UOB-COL-17**GTF2H5**6q25.3Functions in gene transcription and DNA repair**No**Likely to be involved in carcinogenesis [33].UOB-COL-18**DAPK1**9q21.33Calcium/calmodulin-dependent serine/threonine kinase involved in multiple cellular signalling pathways that trigger cell survival, apoptosis, and autophagy**No**DAPK promoter methylation may be involved in NSCLC carcinogenesis [34]; DAPK promoter methylation and abnormal expression of DAPK mRNA in acute leukaemia patients [35]; DAPK as potential therapeutic target [36].^#^ Information obtained from Genecards.og. * Identified in cell line from patient with melanoma by SEREX (Cancer Immunome Database).
biomolecules-12-01058-t002_Table 2Table 2Immunoscreening of antigens with sera from each patient with CRC CLR and HVs indicated that UOB-COL-1 and UOB-COL-7 were only recognised by patient sera.SEREX IDRecognition by CRC-CLR Sera during ImmunoscreeningHV Sero-ScreeningCC005CC010CC014Total12345678Total**UOB-COL-1****+****+****+****3/3****−****−****−****−****−****−****−****−****0/8**UOB-COL-2−++2/3**−**++**−**++++6/8UOB-COL-3−−+1/3++++++++8/8UOB-COL-4+++3/3**−**+++++++7/8UOB-COL-5+−+2/3**−**++**−**++++6/8UOB-COL-6+++3/3**−****−**++**−****−****−****−**2/8**UOB-COL-7****+****+****+****3/3****−****−****−****−****−****−****−****−****0/8**UOB-COL-8++−2/3**−**+++++++7/8UOB-COL-9+−+2/3+++**−**++++7/8UOB-COL-10−++2/3++++++++8/8UOB-COL-11−−+1/3++++++++8/8UOB-COL-12−−+1/3**−**++**−**++++6/8UOB-COL-13−−+1/3++++++++8/8UOB-COL-14−−+1/3++++++++8/8UOB-COL-15−++2/36/1411/1412/148/1411/1411/1412/1411/1412/14UOB-COL-16+ND+2/2
UOB-COL-17++ND2/2
UOB-COL-18+NDND1/1
Total number of antigens recognised by reactive sera10/189/1615/16

ND: not done.
biomolecules-12-01058-t003_Table 3Table 3Analysis of antigen transcription in 100 CRC patients compared with five normal donors using publicly available RNA-seq data. Nine probesets, each including IGHG2 and IGHG3, had >2-fold increased transcription of the heavy chain constant regions of IgG2 and IgG3 in tumour versus normal tissue.Probeset IDGene Symbol*p*-ValueFold-Change211868_x_atIGH///IGHA1///IGHA2///IGHD///IGHG1///IGHG2///**IGHG3**///IGHM///IGHV4-312.13 × 10^−5^2.69211641_x_atIGHA1///IGHA2///IGHD///IGHG1///**IGHG3**///IGHM///IGHV4-310.0012.47211637_x_atIGH///IGHA1///IGHA2///IGHD///IGHG1///**IGHG3**///IGHG4///IGHM///IGHV3-23///IGHV4-310.0013.09214916_x_atIGHA1///IGHA2///IGHG1///**IGHG3**///IGHM///IGHV3-23///IGHV4-310.0022.62211650_x_atIGH///IGHA1///IGHD///IGHG1///**IGHG3**///IGHM///IGHV3-23///IGHV4-310.0022.59211639_x_atIGH///IGHA1///IGHA2///IGHD///IGHG1///**IGHG3**///IGHG4///IGHM///IGHV4-310.0062.09217281_x_atIGH///IGHA1///IGHA2///IGHG1///**IGHG2**///**IGHG3**///IGHM///IGHV4-310.00672.49216557_x_atIGHA1///IGHD///IGHG1///**IGHG3**///IGHM///IGHV4-310.0072.49211635_x_atIGHA1///IGHA2///IGHD///IGHG1///**IGHG3**///IGHG4///IGHM///IGHV4-310.0182.13


### 3.5. IgG3 Protein Was Not Detected in CRC Primary Tumour Cell Lines

To determine whether IgG heavy chain fragments were being expressed by the tumour tissue, in the same way that other epithelial cancers including CRC have been shown to previously [37], or whether antibody recognition of IGHG3 products were due to infiltration of the adaptive immune response into the tumour as part of the CLR reaction, we examined the expression of IgG3 in cell lines and Hansestadt Rostock colorectal cancer (HROC) patient samples. The epithelial-derived SW480 cell line (CVCL_0546) from a patient with Dukes’ B [38] and the epithelial-like HCT116 [39] cell line (CVCL_0291) were used as positive controls for intracellular IgG2 and IgG3 protein production (data not shown). In addition, we examined 20 HROC samples from 18 patients that represented with adenocarcinoma of the colon from a broad range of anatomic sites, grading and staging types, and molecular classes (Appendix A). We predominantly observed single IgG3 positive cells in stroma and parenchyma and weak staining in the stroma of some CRC tissues (Figure 2; Table 4). Despite two of the patients having high lymphocytic stroma reaction and CLR, neither showed high levels of intracellular IgG3 expression.

### 3.6. IgG3 Levels Were Elevated in CRC Patient Sera

We found there was an increase in the levels of IgG3 protein in the sera of patients with CRC when compared to healthy donors (*p* = 0.01; Figure 3). However, the levels of IgG3 in the CLR-CRC patients (CC005 and CC010) used for immunoscreening fell within the range of the CRC group.

### 3.7. IGH Transcription Was Indicative of Clinical Features of CRC

Gene expression analysis of 100 CRC samples in the GSE5206 dataset indicated associations between clinical features and expression of the genes of interest identified in this study (Appendix A). Only columns/rows of data that contained a >two-fold change in gene transcript levels are shown in Table 5, Table 6, Table 7, Table 8, Table 9 and Appendix A. With regard to Dukes’ stage, only three probesets, each IGH family members (211650_x_at; 214916_x_at; 216557_x_at), showed a >two-fold difference in expression, each in Dukes’ stage B down versus in situ (IS). IGH, family members were increased in 0 versus 2 and 0 versus 2B, each representing increased expression in less aggressive stages of CRC and reinforcing the idea that IGH family member transcription decreases with CRC progression, especially in AJCC stage 2 and 2B.

When considering the TNM staging of CRC, in the T stage, RPL37A was increased in 1 versus 3, 4 and X (Table 6 and Appendix A). The N stage showed no two-fold differences, while three probesets (211650_x_at, 216557_x_at, 217281_x_at), each IGH family members, had fold changes that were >two in M stage (2.05, 2.18 and 2.12, respectively) for 0 up versus X. Four probesets (211650_x_at, 216557_x_at, 217281_x_at and 211430_s_at) had fold changes >2 (2.18, 2.06, 2.12 and 2.08, respectively) in 1 up versus X in M stage.

Transcription of IGH family members demonstrated >two-fold increase in colon (ascending) versus colon (NOS), colon ascending (down) versus hepatic flexure, colon (cecum) up versus colon (NOS), colon (cecum) down versus hepatic flexure and colon descending up versus colon (NOS) (Table 7), while a number of IGH probesets were decreased in colon (ascending) down versus colon (splenic flexure), colon (cecum) down versus colon (splenic flexure), colon (descending) down versus colon (splenic flexure) and colon (NOS) down versus colon (sigmoid; splenic flexure and transverse). *p*-values for this data are shown in Appendix A.

With regard to recurrence type, CYB5R3 showed a >two-fold decrease in transcript levels in distance of recurrence to peritoneum or ascites versus distance from site of recurrence, behaviour NOS; versus local recurrence, behaviour NOS; versus never disease free, since diagnosis and versus regional recurrence, behaviour NOS (Table 8). *p*-values for this data are shown in Appendix A. IGH family members probesets were decreased by 9–12-fold in distance of recurrence to peritoneum or ascites versus distance to site of recurrence, behaviour NOS down (13 of 16 IGH probesets) and versus local recurrence (12 of 16 IGH probesets). Two-fold increases in transcription in IGH family member probesets was observed in local recurrence, behaviour NOS up versus never disease free since diagnosis; versus regional recurrence, behaviour NOS and versus regional tissue recurrence of invasive cancer.

### 3.8. Expression of Three CLR CRC Antigens Indicated Overall Survival in CRC Patients

Below median levels of death-associated protein kinase 1 (DAPK1) and above median levels of SH3 domain containing ring finger 2 (SH3RF2) and General transcription factor IIH subunit 5 (GTF2H5) were associated with improved overall survival in the GSE17537 dataset (*n* = 232 CRC patients) [40] (Figure 4). This was not the case for members of the IGH family despite their preponderant association with clinical features.

## 4. Discussion

Through immunoscreening, we identified 18 antigens that were recognised by sera from three CRC patients with CLR; nine were novel and have yet to be identified as transcribed genes through sequence analysis. However, eight of the known antigens (the exception being IGHG2) had been shown to play a role in various tumour types previously. UOB-COL-1, encoding the immunoglobulin heavy constant 3 (G3m marker), had been shown to be increased in inflamed colonic tissue, including ulcerative colitis and Crohn’s disease [41]. Indeed, immunoglobulins have been found to be intrinsically produced by a range of, often epithelial, cancer types [37,42,43,44], having been shown to play a role in cancer initiation, proliferation, invasion, metastasis and survival [45,46,47,48,49]. Immunoglobulins have been shown to be expressed in CRC tissue by a number of groups [42,50], while Geng et al. [51] recently reported that the immunoglobulins expressed by CRC tend to be IgG, rather than other classes of IgH, and show unique VHDJH patterns and somatic hypermutation hotspots. IGHG3 specifically has also been shown to be overexpressed in a number of cancers, again predominantly epithelial tumours, including non-squamous non-small cell lung cancer [24,48], breast cancer [23,52], prostate cancer [22], malignant mesothelioma [21] and CRC [51,53]. Recently Xu et al. [54] showed that mouse IgG could be used to treat and prevent melanoma, breast and colon cancer in syngeneic animal models, while treatment of patients with autoimmunity with anti-IgG was incidentally shown to reduce tumour growth in patients [55,56,57]. The HROC collection of low passage CRC cell lines have been shown to produce IgG previously, and the intrinsic levels are significantly higher than those found in human T cells (used as controls) [58]. However, levels of IgG were very low in the medium, suggesting that although colon cancer cells can generate IgG, unlike B cells, they do not secrete it. We found that few of the HROC tissues we tested expressed notable levels of IgG3, suggesting that the samples we had performed immunoscreening on had in fact demonstrated an infiltration of the tumour by IgG3 secreting B cells rather than intrinsically producing the heavy chain of IgG3. Although Mon et al. showed that IGHG3 expression correlates with non-small cell lung cancer patients who respond to chemotherapy [59] and all three CLR patients tested by SEREX reacted to the IGHG3 gene product, IgG3 protein levels in low passage CRC samples appeared to be very low. Indeed, IgG3 levels in the CRC patient sera were elevated compared with normal donors, again suggesting that the antibody response to IGHG3 products was to infiltrating B cells and/or antibodies rather than IGHG3 production within the tumour cells.

Of the other antigens identified using CLR sera from CRC patients, SH3RF2 was one of two proteins that has been directly linked to CRC through its activity as an oncogene. SH3RF2 has previously been shown to have elevated expression in 159 CRC tissues and to correlate with poor prognostic indicators [32]. In contrast, we found above-median levels of SH3RF2 to correlate with improved overall survival (*p* = 0.02). Although SH3RF2 has been shown to have high expression in colon cancer, with moderate membranous positivity, SH3RF2 is described as having low cancer specificity and is not prognostic. CYB5R3 (identified as UOB-COL-4 in our study) has been shown to be downregulated in breast cancer, negating its ability to detoxify the responsive hydroxylamine metabolites of known mammary carcinogens [60]. Blanke et al. [26] demonstrated that a particular polymorphism in the CYB5R3 gene was associated with breast cancer risk in women. Their study suggested that CYB5R3 may be directly involved in the development of other tumours associated with aromatic and heterocyclic amine exposures, such as CRC. When CYB5R3 siRNA was injected intravenously into mice they showed a decrease in lung cancer burden [26], with cellular effects including signalling alterations associated with extravasation, transforming growth factor beta (TGFβ) and hypoxia-inducible factor 1-alpha (HIFα) pathways, and apoptosis [61].

RPL37A (named UOB-COL-11 in our study) has also been found to be upregulated in high grade astrocytomas [62] and to have a general association with lifetime and overall glioblastoma survival (*p*-value < 0.05 in both cases) [63]. In breast cancer, RPL37A expression, in conjunction with metastases suppressor 1 (MTSS1), as well as SET and MYND domain-containing protein 2 (SMYD2), were shown to predict the response of breast cancer patients to neoadjuvant doxorubicin and cyclophosphamide [31]. We found that above-median levels of RPL37A in CRC patients significantly correlated with improved disease-free survival (*p* = 0.0055), supporting the belief that RPL37A may play a role in cell death, as suggested in glioblastoma [63]. SLC34A2 (named UOB-COL-5 in our study) is a sodium-dependent phosphate transporter that is overexpressed in CRC, and its expression was significantly correlated with N stage in CRC [64]. SLC34A2 can be used to stratify patient prognosis in stage II and III CRC, while high levels of SLC34A2 can be correlated with higher post-operative metastases rates and act as an independent adverse factor affecting patient prognosis. Knock-down experiments showed that the absence of SLC34A2 inhibited cell proliferation, colony formation, induced apoptosis and arrested the cell cycle, while in in vivo studies, SLC34A2 knock-down prevented xenograft growth and promoted apoptosis.

DAPK1, identified as UOB-COL-18 in this study, is known to be an important serine/threonine kinase that acts as a positive regulator of programmed cell death and is also involved in the regulation of autophagy and cell migration [36]. The DAPK promoter has been reported to be methylated in several types of cancer [34,65,66], thus disrupting the process of programmed cell death. However, Satoh et al. [65] have demonstrated that in CRC, methylation alone is not responsible for silencing DAPK and histone deacetylation is also required. Inhibition of histone deacetylation acts synergistically with the inhibition of DNA methylation to induce DAPK gene expression. Therefore, DAPK1 may be a suitable target for the treatment of CRC patients through the activation of apoptosis using histone deacetylase inhibitors, with the frequency of DAPK1 expression dictating feasibility. Models of colon cancer [67] have shown that the loss of DAPK1 expression has been shown to impact migratory capacity [68], tumour cell dissemination and increase invasiveness. We found that above-median levels of DAPK1 were associated with poorer overall survival (*p* = 0.005), which supports its potential as a target for CRC therapy.

Zinc finger 465 (ZNF465; named UOB-COL-3 in this study) has previously been identified through the autologous immunoscreening of a library made from a patient with acute myeloid leukaemia (AML) [25]. ZNF465 is a Krüppel-type zinc finger transmembrane protein with a 5′ Krüppel-associated box domain that is typical of negative regulators of gene transcription [69]. We found ZNF465 to be expressed in AML and chronic myeloid leukaemia but not HVs. To date, most Krüppel-type zinc finger proteins are found to be transcriptional repressor proteins involved in a variety of biological processes [69,70]. General Transcription Factor IIH, polypeptide 5 (GTF2H5), is a small protein that stabilizes the multi-subunit transcription repair factor IIH (TFIIH). TFIIH plays a key role in a nuclear excision repair in healthy tissues and was identified as one of six most significantly mutated genes in squamous cell carcinoma related to carcinogenesis [33]. Above-median levels of GTF2H5 have been associated with improved overall survival and reduced disease-free survival, with a much higher level of significance in its inverse association with disease free survival. This may be explained by its role in nuclear excision repair, whereby its greater expression leads to genetic repair and removes neo-antigens that would otherwise act as immune targets following chemotherapy and/or surgery.

The limitations of this study included the small number of sera samples from CRC CLR patients that were used for immunoscreening. Only IGHG2 and IGHG3 had elevated expression in CRC patients compared with healthy donors, as determined by RNA-seq analysis in an independent dataset, but protein levels were not determined as tissue was not available within the patient samples we had collected and intrinsic IGHG3 expression was not found in the CRC primary cell lines. There is not a linear relationship between RNA transcription and protein levels in higher eukaryotes [71] due to various biological and technical factors, including codon bias, RNA secondary structure, as well as protein abundance and turnover; however, the presence of the proteins in patients samples would have provided confirmation of their presence, even though most did not correlate with disease (except IGHG2 and IGHG3) or overall survival (except SHR3RF2, GTF2H5 and DAPK1).

The aim of this study was to determine which antigens were recognised by patients with CRC CLR and whether these same antigens could provide insights into why some patients respond well to treatment and others do not. In addition, we were hoping to find novel targets for future treatments of CRC patients, many of whom lack MSI and fail to respond to ICI treatment. We identified two antigens that were recognised by all three CRC CLR patients. Although one of the antigens was novel (UOB-COL-7) with unknown function, the other, IGHG3, showed a number of associations with the clinical features of CRC in a large dataset study. Future studies will determine which antigen(s) are being recognised by these CRC-associated antibodies and whether blocking these antibodies could reduce tumour growth as described previously [53]. Of note was the identification of DAPK1, SH3RF2 and GTF2H5, whose expression provides a clear indication of overall survival and targets worthy of further investigation for CRC treatment.

## Figures and Tables

**Figure 1 biomolecules-12-01058-f001:**
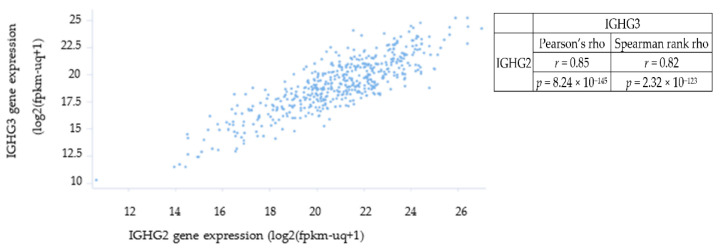
IGHG2 and IGHG3 expression levels were examined in 571 CRC patient samples following RNA-seq (TCGA-COAD dataset). There was no evidence of a unique sub-population of patients with elevated IGHG3 expression within the CRC population. However, we did find a significant association between the levels of IGHG2 (*x*-axis) and IGHG3 (*y*-axis) expression suggesting the transcription of these constant regions have similar expression profiles in CRC patients (*p* = 8.25 × 10^−145^).

**Figure 2 biomolecules-12-01058-f002:**
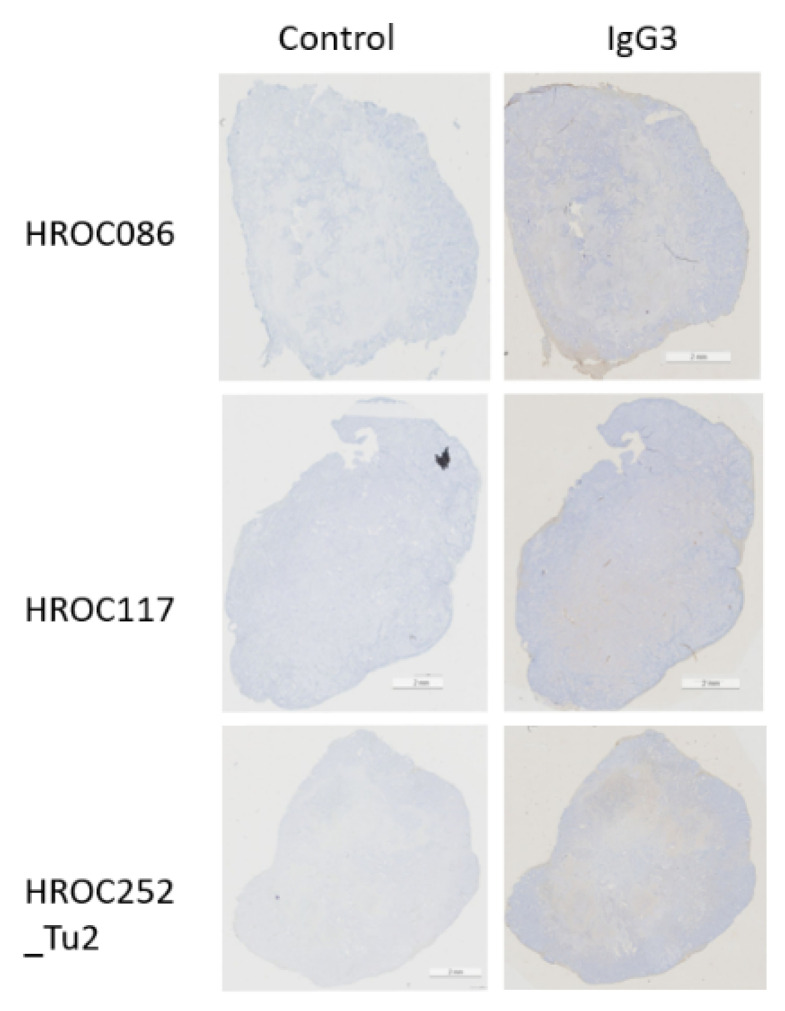
Immunolabelling of intracellular IgG3 in CRC samples. Examples of staining include single cells, strong in stroma (HROC086) and parenchyma (HROC0252_Tu2) and very slight background staining (HROC117).

**Figure 3 biomolecules-12-01058-f003:**
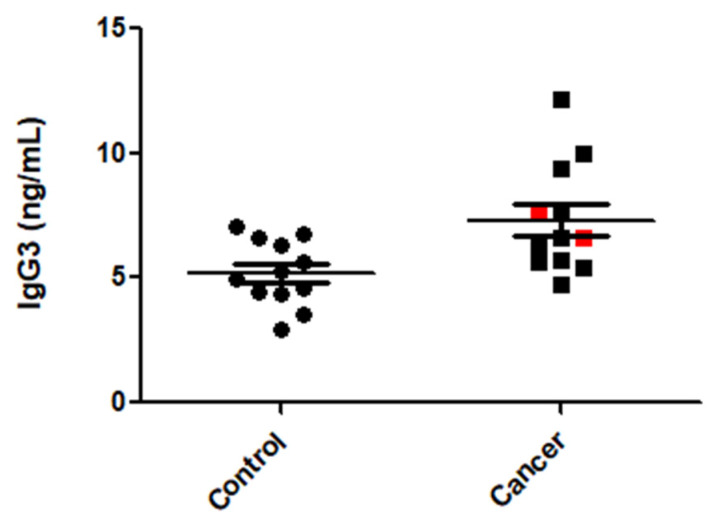
IgG3 levels in CRC patients. Levels of IgG3 were elevated in the sera of CRC patients used for immunoscreening when compared to healthy donors (*p* = 0.01), however within this group the IgG3 sera levels in CC005 and CC010 (shown as red squares) were not elevated above the rest of the group of CRC patients.

**Figure 4 biomolecules-12-01058-f004:**
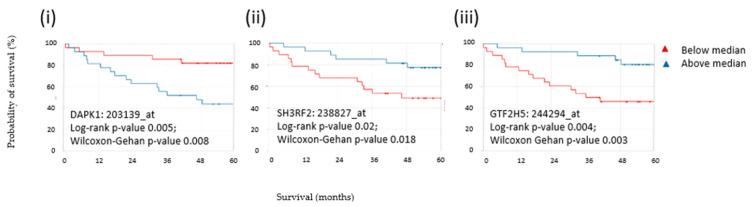
Gene expression analysis was used to determine the impact of above and below median antigen expression on survival during follow-up. The probability of overall survival is shown on the *y*-axis, while time in months is shown on the *x*-axis. Below median levels of (**i**) DAPK1 and above median levels of (**ii**) SH3RF2 and (**iii**) GTF2H5 were associated with improved 5-year overall survival in CRC patients.

**Table 4 biomolecules-12-01058-t004:** IgG3 expression in CRC samples.

Sample ID	Molecular Subtype	Control	IgG3	Observations Regarding IgG3 Staining
HROC40	CIMP-H, non MSI	−	−/+	Very slight background staining
HROC54	CIMP-H, non MSI	−	+	Very slight background staining
HROC60	CIMP-H, non MSI	−	+	Very slight background staining
HROC62	spStd	−	−	Diffusely in necrosis
HROC86	spStd	−	++	Single cells strong in stroma
HROC117	CIMP-H, non MSI	−	++	Very slight background staining
HROC126	spStd	−	−	Weak and diffusely in necrosis
HROC131	spMSI-H	−	−	
HROC155	spStd	−	−	Single cells moderate in stroma
HROC159	spMSI-H	−	+	Moderate staining in stroma
HROC169	CIMP-H, non MSI	−	−	
HROC212	spMSI-H	−	+	Single cells in stroma
HROC252_Tu1	Lynch	−	+	Diffusely IgG3 in stroma
HROC252_Tu2		−	++	Single cells in stroma, also in parenchyma
HROC252_Tu3		−	+	Single cells in stroma, low in parenchyma
HROC257	spMSI-H	−	+	Weak and diffusely IgG3 in necrosis, weak in stroma
HROC260	spStd	−	−	Single cells weak in stroma
HROC269	spMSI-H	−	−	Weak in necrosis
HROC315	Lynch	−	+	Necrotic tumour, IgG3 strong in single cells in stroma
HROC324	Lynch	−	−	Weak in necrosis
Tonsil		−	+++/++++	Regions of heavily brown staining

CIMP-H: CpG island methylator phenotype-high; Lynch: Lynch syndrome, also known as hereditary non-polyposis colorectal cancer (HNPCC); MSI: microsatellite instability; sp: sporadic; spStd: sporadic standard type.

**Table 5 biomolecules-12-01058-t005:** Fold change in antigen transcript levels that achieved significance, when comparing AJCC stage. Light green cells indicate values that are increased by greater than 2-fold, yellow cells indicate values that are decreased by greater than 2-fold.

Probeset ID	Gene Symbol	Stage 0 down vs. Stage				Stage 2B down vs. Stage 3B
		2	2B	3B	4	
213459_at	**RPL37A**	1.42	2.41	1.04	1.27	−2.31
211650_x_at	**IGH family members**	2.25	2.73	1.77	1.89	−1.54
214916_x_at		2.19	2.86	1.63	2.05	−1.75
216557_x_at		2.14	2.11	1.60	1.77	−1.32
217281_x_at		1.91	2.04	1.50	1.60	−1.36

**Table 6 biomolecules-12-01058-t006:** Fold change in antigen transcript levels that achieved significance, when comparing T stage. Light green cells indicate values that are increased by greater than 2-fold, yellow cells indicate values that are decreased by greater than 2-fold.

Probeset ID	Gene Symbol	Stage Comparisons				
		1 vs. 3	1 vs. 4	1 vs. IS	1 vs. X	3 vs. IS
213459_at	**RPL37A**	2.57	2.38	1.77	2.52	−1.45
211639_x_at	**IGH family members**	−1.24	−1.26	−2.20	−1.32	−1.77
211650_x_at		1.13	1.01	−1.78	1.04	−2.00
214916_x_at		1.16	1.11	−1.77	−1.01	−2.04

**Table 7 biomolecules-12-01058-t007:** Fold change in antigen transcript levels that achieved significance, when comparing collection sites (colon ascending, descending and nos). Light green cells indicate values that are increased by greater than 2-fold, yellow cells indicate values that are decreased by greater than 2-fold.

Probeset ID	Gene Symbol	COLON (ASCENDING) vs. COLON (NOS)	COLON (ASCENDING)		COLON (DESCENDING)		COLON (DESCENDING)		COLON (NOS)				
			vs. COLON (SPLENIC FLEXURE)	vs. HEPATIC FLEXURE	vs. COLON (NOS)	vs. RECTUM (NOS)	vs. COLON (SPLENIC FLEXURE)	vs. HEPATIC FLEXURE	vs. COLON (SIGMOID)	vs. COLON (SPLENIC FLEXURE)	vs COLON (TRANSVERSE)	vs. HEPATIC FLEXURE	vs. RECTOSIGMOID JUNCTION
211430_s_at	**IGH family members**	**2.26**	−1.81	−1.2	1.29	−2.06	−3.19	−2.11	−2.38	−4.1	−1.5	−2.71	−2.00
211635_x_at		1.91	−2.17	1.16	1.26	−1.64	−3.3	−1.31	−2.37	−4.15	−1.56	−1.65	−1.46
211637_x_at		4.23	−2.56	2.14	2.89	−1.63	−3.75	1.46	−3.86	−10.8	−3.6	−1.98	−2.74
211639_x_at		2.62	−1.94	1.66	2.2	−1.37	−2.31	1.39	−2.86	−5.08	−2.46	−1.58	−2.08
211641_x_at		1.62	−1.87	1.82	1.38	−1.27	−2.19	1.55	−1.59	−3.03	−1.37	1.12	−1.18
211650_x_at		3.04	−2.13	2.27	2.53	−1.5	−2.56	1.89	−3.01	−6.47	−2.58	−1.34	−2.26
211868_x_at		2.5	−1.54	1.43	1.91	−1.35	−2.02	1.09	−2.52	−3.86	−2.26	−1.75	−2.20
214916_x_at		2.8	−1.96	2.76	1.98	−1.76	−2.78	1.95	−2.73	−5.5	−2.16	−1.01	−1.92
216557_x_at		3.68	−2.22	2.34	2.26	−1.86	−3.61	1.44	−3.31	−8.15	−2.93	−1.57	−2.47
217236_x_at		1.33	−1.67	1.97	1.16	−1.19	−1.91	1.72	−1.31	−2.22	−1.19	1.48	−1.12
217281_x_at		2.95	−2.12	2.56	1.98	−1.69	−3.15	1.72	−2.81	−6.26	−2.3	−1.15	−2.23
217360_x_at		1.33	−1.75	1.33	1.34	−1.09	−1.73	1.34	−1.4	−2.33	−1.21	1.0	−1.16

**Table 8 biomolecules-12-01058-t008:** Fold change in antigen transcript levels that achieved significance, when comparing collection sites including colon (cecum, nos, splenic flexure, sigmoid, transverse) and rectum (nos). Light green cells indicate values that are increased by greater than 2-fold, yellow cells indicate values that are decreased by greater than 2-fold.

Probeset ID	Gene Symbol	COLON (CECUM) vs. COLON (NOS)	COLON (CECUM)	COLON (CECUM) vs. COLON (NOS)	COLON (NOS) vs. RECTUM (NOS)	COLON (SIGMOID)		COLON (SPLENIC FLEXURE)				COLON (TRANSVERSE) vs. HEPATIC FLEXURE	HEPATIC FLEXURE vs. RECTUM (NOS)
			vs. COLON (SPLENIC FLEXURE)			vs. COLON (SPLENIC FLEXURE)	vs. HEPATIC FLEXURE	vs. COLON (TRANSVERSE)	vs. HEPATIC FLEXURE	vs. RECTOSIGMOID JUNCTION	vs. RECTUM (NOS)		
211430_s_at	**IGH family members**	**2.46**	−1.67	2.46	−2.65	−1.72	−1.14	2.74	1.51	2.05	1.55	−1.81	1.02
211635_x_at		1.95	−2.13	1.95	−2.06	−1.75	1.44	2.67	2.51	2.84	2.02	−1.06	−1.25
211637_x_at		3.85	−2.81	3.85	−4.71	−2.81	1.95	3.01	5.47	3.95	2.30	1.82	−2.38
211639_x_at		2.66	−1.91	2.66	−3.02	−1.78	1.81	2.07	3.22	2.45	1.68	1.55	−1.91
211641_x_at		1.49	−2.03	1.49	−1.75	−1.90	1.78	2.21	3.40	2.56	1.73	1.54	−1.96
211650_x_at		2.87	−2.25	2.87	−3.79	−2.15	2.26	2.51	4.85	2.87	1.71	1.93	−2.84
211868_x_at		2.29	−1.69	2.29	−2.58	−1.53	1.44	1.70	2.20	1.75	1.50	1.29	−1.47
214916_x_at		2.34	−2.35	2.34	−3.49	−2.02	2.69	2.55	5.43	2.87	1.58	2.12	−3.44
216557_x_at		3.29	−2.48	3.29	−4.20	−2.46	2.11	2.79	5.19	3.31	1.94	1.86	−2.68
217236_x_at		1.28	−1.73	1.28	−1.39	−1.69	1.94	1.87	3.29	1.97	1.60	1.76	−2.06
217281_x_at		2.75	−2.27	2.75	−3.36	−2.23	2.44	2.71	5.43	2.80	1.86	2.00	−2.92
217360_x_at		1.33	−1.75	1.33	−1.47	−1.67	1.40	1.93	2.33	2.01	1.58	1.21	−1.47

**Table 9 biomolecules-12-01058-t009:** Fold change in antigen transcript levels that achieved significance, when comparing recurrence type. Light green cells indicate values that are increased by greater than 2-fold, yellow cells indicate values that are decreased by greater than 2-fold.

Probset ID	Gene Symbol	DIST RECUR TO PERITONEUM OR ASCITES					DIST SITE OF RECUR, BEHAV NOS (40)	DIST SITE OF RECUR, BEHAV NOS (40)		LOCAL RECURRENCE, BEHAV NOS (10)			NEVER DISEASE FREE SINCE DX (70)	REGIONAL RECURRENCE, BEHAV NOS (20)
		vs. DIST SITE OF RECUR, BEHAV NOS (40)	vs. LOCAL RECURRENCE, BEHAV NOS (10)	vs. NEVER DISEASE FREE SINCE DX (70)	vs. REGIONAL RECURRENCE, BEHAV NOS (20)	vs. REGIONAL TISSUE RECUR OF INVAS CA (21)	vs LOCAL RECURRENCE, BEHAV NOS (10)	vs REGIONAL RECURRENCE, BEHAV NOS (20)	vs REGIONAL TISSUE RECUR OF INVAS CA (21)	vs NEVER DISEASE FREE SINCE DX (70)	vs REGIONAL RECURRENCE, BEHAV NOS (20)	vs REGIONAL TISSUE RECUR OF INVAS CA (21)	vs REGIONAL TISSUE RECUR OF INVAS CA (21)	vs REGIONAL TISSUE RECUR OF INVAS CA (21)
1554574_a_at	**CYB5R3**	−2.31	−2.78	−2.58	−2.02	−1.85	−1.2	1.15	1.25	1.08	1.38	1.5	1.39	1.09
211430_s_at	**IGH family members**	1.06	−2	1.38	−1.38	3.2	−2.11	−1.46	3.03	2.76	1.45	6.4	2.32	4.42
211635_x_at		−1.02	−10.57	−1.32	−1.61	−2.26	−10.35	−1.58	−2.21	7.99	6.57	4.68	−1.7	−1.4
211637_x_at		−1.39	−10.91	−1.92	−2.70	−2.02	−7.83	−1.94	−1.45	5.69	4.03	5.4	−1.05	1.34
211639_x_at		−1.09	−3.63	−1.25	−1.73	−1.47	−3.31	−1.58	−1.34	2.9	2.1	2.47	−1.17	1.18
211641_x_at		−1.26	−4.86	−1.34	−1.74	−1.59	−3.86	−1.38	−1.26	3.62	2.8	3.06	−1.18	1.09
211650_x_at		−1.39	−9.01	−1.45	−2.01	−2.01	−6.48	−1.44	−1.45	6.2	4.49	4.48	−1.38	−1.00
211868_x_at		1.20	−2.47	1.09	−1.25	1.08	−2.97	−1.51	−1.12	2.69	1.97	2.66	−1.01	1.35
214916_x_at		−1.2	−11.5	−1.40	−1.84	−1.88	−9.58	−1.53	−1.57	8.19	6.26	6.1	−1.34	−1.03
216557_x_at		−1.47	−12.9	−1.67	−2.55	−1.82	−8.38	−1.74	−1.24	7.36	4.82	6.75	−1.09	1.4
217236_x_at		−1.3	−2.13	−1.43	−1.59	−1.71	−1.63	−1.22	−1.31	1.49	1.33	1.24	−1.2	−1.07
217281_x_at		−1.18	−10.01	−1.32	−1.99	−1.82	−8.5	−1.69	−1.55	7.59	5.04	5.49	−1.38	1.09
217360_x_at		−1.17	−3.25	−1.06	−1.57	−1.75	−2.78	−1.34	−1.49	3.07	2.07	1.86	−1.65	−1.11
234419_x_at		−1.02	−2.24	−1.04	−1.24	−1.6	−2.21	−1.22	−1.58	2.15	1.8	1.4	−1.54	−1.29

## Data Availability

All data supporting reported results can be found in the Appendix A.

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
