# Peer review of "Identification of the Antigens Recognised by Colorectal Cancer Patients Using Sera from Patients Who Exhibit a Crohn’s-like Lymphoid Reaction"

_biomolecules, 2022, doi:10.3390/biom12081058_

Round 1

Reviewer 1 Report

In this original article, the authors identified 18 antigens that were recognized by sera from CRC patients with CLR by immunoscreening. Nine of them were novel and have yet to be identified as transcribed genes through sequence analysis. Eight of the known antigens had been shown to play a role in various tumor types previously. They identified two antigens that were recognized by all 3 CRC-CLR patients. Although one of the antigens was novel (UOB-COL-7) with an unknown function, the other, IGHG3, showed a number of associations with the clinical features of CRC in a large dataset study. Of note was the identification of DAPK1, SH3RF2, and GTF2H5, whose expression provides a clear indication of overall survival.

This study is well designed and well presented. The used methods are properly described, and the results are clear and well established. The results are discussed in a moderate and logical manner, the new results are interpreted in the light of the international literature.

Minor comments:

- instead of tables containing too much data, data – where possible- should be represented graphically.

- A few minor mistakes need to be corrected:

line 109 and line 286: Duke’s B – Dukes’ B

line 406: over expressed - overexpressed.

I suggest minor revision. 

Author Response

Thank you for your kind comments.

Response to Minor comments:

instead of tables containing too much data, data – where possible- should be represented graphically.

We agree. We could change Tables 5a-d to show only the percentage of probes whose expression was increased or decreased in each comparison but don't think that would be very meaningful. However we did remove all probes that had no >2 fold change in expression in the comparisons to try to minimise the size of the Tables.

- A few minor mistakes need to be corrected:

line 109 and line 286: Duke’s B – Dukes’ B

Corrected.

line 406: over expressed - overexpressed.

Corrected

I suggest minor revision. 

Thank you

Reviewer 2 Report

This is an interesting study that aimed to identify the antigens recognised by colorectal cancer patients with a Crohn’s like lymphoid reaction. Please, check throughout the text for spelling errors and consistency of abbreviations. In the Abstract please check the word count according to the guidelines and state more clearly the aim of the study. Please, if possible, try to improve quality of Figure 1. Limitations of the study should be stated more clearly in the discussion section. I would also suggest to add more discussion on future perspectives and potential clinical implications of the findings.

Author Response

This is an interesting study that aimed to identify the antigens recognised by colorectal cancer patients with a Crohn’s like lymphoid reaction. 

Thank you

Please, check throughout the text for spelling errors and consistency of abbreviations.

Checked.

In the Abstract please check the word count according to the guidelines and state more clearly the aim of the study.

Word count now 200 words and we have made the aims (lines 21-22 clearer). 'To identify the immune targets recognised by CRC-CLR patient sera..... '

Please, if possible, try to improve quality of Figure 1.

We have made the x axis longer, increased the font on the labels and moved the Table showing statistical information to the right. Hopefully makes viewing the data easier.

Limitations of the study should be stated more clearly in the discussion section.

Detailed in lines 465 - 475 as follows 'The limitations of this study included the small number of sera samples from CRC CLR patients that were used for immunoscreening. Only IGHG2 and IGHG3 had elevated expression in CRC patients compared with healthy donors as determined by RNA-seq analysis in an independent dataset, but protein levels were not determined as tissue was not available within the patient samples we had collected and intrinsic IGHG3 expression was not found in the CRC primary cell lines. There is not a linear relationship between RNA transcription and protein levels in higher eukaryotes [65] due to various biological and technical factors including codon bias, RNA secondary structure as well as protein abundance and turnover but the presence of the proteins in patients samples would have provided confirmation of their presence, even though most did not correlate with disease (except IGHG2 and IGHG3) or overall survival (except SHR3RF2, GTF2H5 and DAPK1).' 

I would also suggest to add more discussion on future perspectives and potential clinical implications of the findings.

Line 483-485 includes future perspective and potential clinical implications of the findings as follows:- Future studies will determine which antigen(s) are being recognised by these CRC-associated antibodies and whether blocking these antibodies could reduce tumour growth as described previously [40]. 

Reviewer 3 Report

Major comments:

1.     Please show the gene name of UOB-COL-6~10; and 12~15 in Table 1.

2.     You described 14 and 4 were identified and confirmed with CC014 and CC005, respectively in Section 3.2 (Line 217-218). However, there are actually 15 and 9 in Table 2. In addition, there should be 15/18 in CC014 recognition. Please check again.

3.     If DAPK1, GTF2H5, and RPL37A are not significantly different expression between CRC and HV in Section 3.4 (Line 237-239), how to elucidate Figure 4 data. And why DAPK1, SH3RF2, and RPL37A should be investigated furthermore in Figure 4.

4.     You mention there is “no expression of IgG3 in primary CRC cell lines” in line 31~32, but there is no cell line data in this manuscript.  

5.     You mention you have enlarged and shown scale bar in Figure 1, but Figure 1 is still the same as the previous version.

6.     What is ICC in the Abstract (line 31).

7.     This study is going to search for antigens recognized by CRC-CLR, but IgG3 was not increased in CRC-CLR compared to CRC patients in Figure 3. It is inconsistent with the Title.

Author Response

  1. Please show the gene name of UOB-COL-6~10; and 12~15 in Table 1. Response: Table 1 only shows the nine previously identified antigens. UOB-COL-6-10 and 12-15 are sequences that do not map to any known gene, but do map to the human genome. Are you happy for us to leave this Table as is?
  2. You described 14 and 4 were identified and confirmed with CC014 and CC005, respectively in Section 3.2 (Line 217-218). However, there are actually 15 and 9 in Table 2. In addition, there should be 15/18 in CC014 recognition. Please check again.   Response: Text in 3.2 and numbers in Table 2 now corrected. Thank you.
  3. If DAPK1, GTF2H5, and RPL37A are not significantly different expression between CRC and HV in Section 3.4 (Line 237-239), how to elucidate Figure 4 data. And why DAPK1, SH3RF2, and RPL37A should be investigated furthermore in Figure 4. Response: When we looked at DAPK1, GTF2H5 and RPL37A expression levels between CRC and HV we found no difference but when we examined above and below median levels of each gene in CRC patients, then there was a correlation with overall survival.
  4. You mention there is “no expression of IgG3 in primary CRC cell lines” in line 31~32, but there is no cell line data in this manuscript.  Response: We examined IgG3 expression in 2 CRC ell lines (SW480 and Hct116; data not shown) and 20 Hansestadt Rostock colorectal cancer (HROC) samples from 18 patients (lines 269-276). Although established from fresh tumour samples these are primary cell lines that have been cultured ex vivo (lines.
  5. You mention you have enlarged and shown scale bar in Figure 1, but Figure 1 is still the same as the previous version. Response: enlarged now. Apologies.
  6. What is ICC in the Abstract (line 31). Response: This has now been removed as part of the shortening of the abstract at the request of another reviewer. IHC on line 104 has been defined.
  7. This study is going to search for antigens recognized by CRC-CLR, but IgG3 was not increased in CRC-CLR compared to CRC patients in Figure 3. It is inconsistent with the Title. Response: Article title has been changed to more accurately reflect the article. Thank you for your insightful comments.

Reviewer 4 Report

Boncheva et al. aimed to identify the antigens the are recognized by patients with CRC-CLR. Further, the author wanted to investigate, if the identified antigen can act as a marker to determine treatment response. The authors have identified two antigens “UOB-COL-7 and IGHG3”. IGHG3 has been shown to be associated with clinical features, including T stage, collection site, and recurrence type. Moreover, the three identified antigens, DAPK1, SH3RF2, and GTF2H5 are shown to be indicators of overall survival in primary CRC samples. Overall, the article is well written, and provide novel information to the readers.

Author Response

Boncheva et al. aimed to identify the antigens the are recognized by patients with CRC-CLR. Further, the author wanted to investigate, if the identified antigen can act as a marker to determine treatment response. The authors have identified two antigens “UOB-COL-7 and IGHG3”. IGHG3 has been shown to be associated with clinical features, including T stage, collection site, and recurrence type. Moreover, the three identified antigens, DAPK1, SH3RF2, and GTF2H5 are shown to be indicators of overall survival in primary CRC samples. Overall, the article is well written, and provide novel information to the readers.

Thank you.

Round 2

Reviewer 3 Report

No more questions.